# Cell type-specific long-range connections of basal forebrain circuit

**Johnny Phong Do[1][†], Min Xu[1][†], Seung-Hee Lee[1][†][‡], Wei-Cheng Chang[1], Siyu Zhang[1], Shinjae Chung[1], Tyler J Yung[1], Jiang Lan Fan[1], Kazunari Miyamichi[2][§], Liqun Luo[2], Yang Dan[1]***

[1]Division of Neurobiology, Department of Molecular and Cell Biology, Helen Wills Neuroscience Institute, Howard Hughes Medical Institute, University of California, Berkeley, United States; [2]Department of Biology, Howard Hughes Medical Institute, Stanford University, Stanford, United States

**Abstract** The basal forebrain (BF) plays key roles in multiple brain functions, including sleep-wake regulation, attention, and learning/memory, but the long-range connections mediating these functions remain poorly characterized. Here we performed whole-brain mapping of both inputs and outputs of four BF cell types – cholinergic, glutamatergic, and parvalbumin-positive (PV+) and somatostatin-positive (SOM+) GABAergic neurons – in the mouse brain. Using rabies virus - mediated monosynaptic retrograde tracing to label the inputs and adeno-associated virus to trace axonal projections, we identified numerous brain areas connected to the BF. The inputs to different cell types were qualitatively similar, but the output projections showed marked differences. The connections to glutamatergic and SOM+ neurons were strongly reciprocal, while those to cholinergic and PV+ neurons were more unidirectional. These results reveal the long-range wiring diagram of the BF circuit with highly convergent inputs and divergent outputs and point to both functional commonality and specialization of different BF cell types.

*For correspondence: ydan@berkeley.edu

[†]These authors contributed equally to this work

Present address: [‡]Korea Advanced Institute of Science and Technology, Republic of Korea; [§]Department of Applied Biological Chemistry, Graduate School of Agricultural and Life Sciences, The University of Tokyo, Tokyo, Japan

## Introduction

The BF has been implicated in a variety of brain functions such as arousal, attention, and plasticity (*Bakin and Weinberger, 1996*; *Brown et al., 2012*; *Everitt and Robbins, 1997*; *Froemke et al., 2013*; *Hasselmo and Sarter, 2011*; *Jones, 2011*; *Lin et al., 2015*; *Saper et al., 2010*; *Sarter et al., 2001*). The dysfunction or loss of BF cholinergic neurons is an important feature of Alzheimer's disease associated with cognitive impairment (*Schliebs and Arendt, 2011*; *Whitehouse et al., 1982*). In addition to forming extensive local synapses (*Xu et al., 2015*; *Yang et al., 2014*; *Zaborszky and Duque, 2000*), BF neurons receive inputs (*Asanuma, 1989*; *Freund and Meskenaite, 1992*; *Grove, 1988a*, *Henny and Jones, 2006*; *Manns et al., 2001*; *Parent et al., 1988*; *Rye et al., 1984*; *Semba et al., 1988*; *Zaborszky and Cullinan, 1992*) and send outputs (*Cullinan and Zaborszky, 1991*; *Grove, 1988b*; *Jones and Cuello, 1989*; *Mesulam and Mufson, 1984*; *Paré and Smith, 1994*) to many other brain areas (*Steriade and McCarley, 2005*; *Zaborszky et al., 2012*). However, how these long-range connections contribute to BF functions remains unclear.

An important challenge in understanding the function of the BF circuit is its neuronal heterogeneity. There are three major cell types spatially intermingled in the BF: cholinergic, glutamatergic, and GABAergic (*Semba, 2000*; *Zaborszky et al., 2012*). Selective lesion or pharmacological manipulation of the cholinergic system is well known to affect multiple brain functions (*Wenk, 1997*). For example, 192-IgG-saporin-mediated lesion of cholinergic neurons impaired the ability of rats to discriminate between signal and non-signal visual events in an attention task (*McGaughy et al., 1996*) and disrupted training-induced cortical map reorganization associated with motor learning

(*Conner et al., 2003*). Glutamatergic and GABAergic BF neurons are also likely to serve important functions (*Lin et al., 2015*). For example, in recent studies the activity of non-cholinergic BF neurons was found to correlate with sustained attention (*Hangya et al., 2015*) or to encode reward and motivational salience information (*Avila and Lin, 2014*; *Lin and Nicolelis, 2008*; *Nguyen and Lin, 2014*), and optogenetic activation of PV+ GABAergic neurons was shown to regulate cortical gamma oscillations (*Kim et al., 2015*). In a study on sleep-wake control, cholinergic, glutamatergic, and PV+ neuron activity was found to promote wakefulness, while SOM+ neurons promoted sleep; these four cell types form extensive but highly specific local connections with each other for brain-state regulation (*Xu et al., 2015*). Thus, to understand the BF circuit function, it is crucial to map its inputs and outputs with cell-type specificity.

Most of the previous studies of BF long-range connections focused on specific regions connected to the BF, making it difficult to assess their whole-brain distribution. Recent advances in virus-assisted circuit tracing (*Callaway and Luo, 2015*; *Huang and Zeng, 2013*) and high-throughput imaging have greatly facilitated whole-brain mapping of long-range connectivity in a cell-type-specific manner (*Oh et al., 2014*; *Osten and Margrie, 2013*). In this study, we traced the long-range inputs and outputs of four genetically defined BF cell types. While the input distributions were similar across cell types, their output patterns showed striking differences. Our quantitative analysis of the whole-brain distributions of inputs and outputs for each BF cell type can serve as an anatomical blueprint for future studies of inter-regional pathways mediating BF functions.

## Results

Four Cre mouse lines were used to target different BF subpopulations for virus-mediated circuit tracing: choline acetyltransferase (ChAT)-Cre for cholinergic neurons, vesicular glutamate transporter 2 (VGLUT2)-Cre for glutamatergic neurons, and PV-Cre and SOM-Cre mice for two subtypes of GABAergic neurons. These four Cre lines have been shown to label largely non-overlapping BF neuron populations with high specificity (*Xu et al., 2015*).

To identify the long-range inputs to each cell type, we used RV-mediated transsynaptic retrograde tracing, which has been shown to label monosynaptic inputs to selected starter cells with high specificity (*Miyamichi et al., 2011*; *Wall et al., 2013*; *Watabe-Uchida et al., 2012*; *Wickersham et al., 2007*). First, we expressed avian-specific retroviral receptor (TVA), enhanced green fluorescent protein (eGFP), and rabies glycoprotein (RG) specifically in each cell type by injecting two Cre-inducible AAV vectors (AAV2-EF1α-FLEX-eGFP-2a-TVA and AAV2-EF1α-FLEX-RG) into the BF of ChAT-, VGLUT2-, PV-, or SOM-Cre mice (*Figure 1A*). The expression of RG was highly cell type specific and not detected in wild-type mice not expressing Cre recombinase (*Figure 1—figure supplement 1*). Two to three weeks later, we injected a modified RV (rabiesΔG-tdTomato+EnvA) that only infects cells expressing TVA, requires RG to spread retrogradely to presynaptic cells (*Figure 1—figure supplement 2*), and contains the tdTomato transgene. After histological sectioning and fluorescence imaging, each sample was aligned to a reference atlas (Allen Mouse Brain Atlas, see Materials and methods) to facilitate 3D whole-brain visualization and quantitative comparison across brain samples (*Figure 1C*). The starter cells (expressing both tdTomato and eGFP) and the transsynaptically labeled presynaptic neurons (expressing tdTomato only) were identified manually, and their locations were registered in the reference atlas (*Figure 1—figure supplement 3*).

Brain samples were excluded from the analyses if very few input neurons (<200) were labeled in the whole brain. As noted in previous studies, due to the extremely efficient interaction between TVA and EnvA-pseudotyped rabies virus, the very low-level expression of TVA in non-Cre-expressing cells (not detectable based on fluorescent protein markers) allows the rabies virus to infect and label these cells with tdTomato at the injection site, independent of synaptic connections with starter cells (*Beier et al., 2015*; *Menegas et al., 2015*; *Miyamichi et al., 2013*; *Ogawa et al., 2014*; *Pollak Dorocic et al., 2014*; *Wall et al., 2013*; *Watabe-Uchida et al., 2012*; *Weissbourd et al., 2014*). However, this local contamination does not compromise the mapping of long-range inputs because RG (required for transsynaptic spread of RV) is not expressed in any non-Cre-expressing cells at sufficient levels for trans-complementation of rabiesΔG to allow transsynaptic spread of RV (*Callaway and Luo, 2015*; *Miyamichi et al., 2013*). To determine the spatial extent of the local contamination, we performed control experiments in the absence of RG and found very few non-specific labeling at >850 µm from the injection center (*Figure 1—figure supplement 2D*). Thus, presynaptic

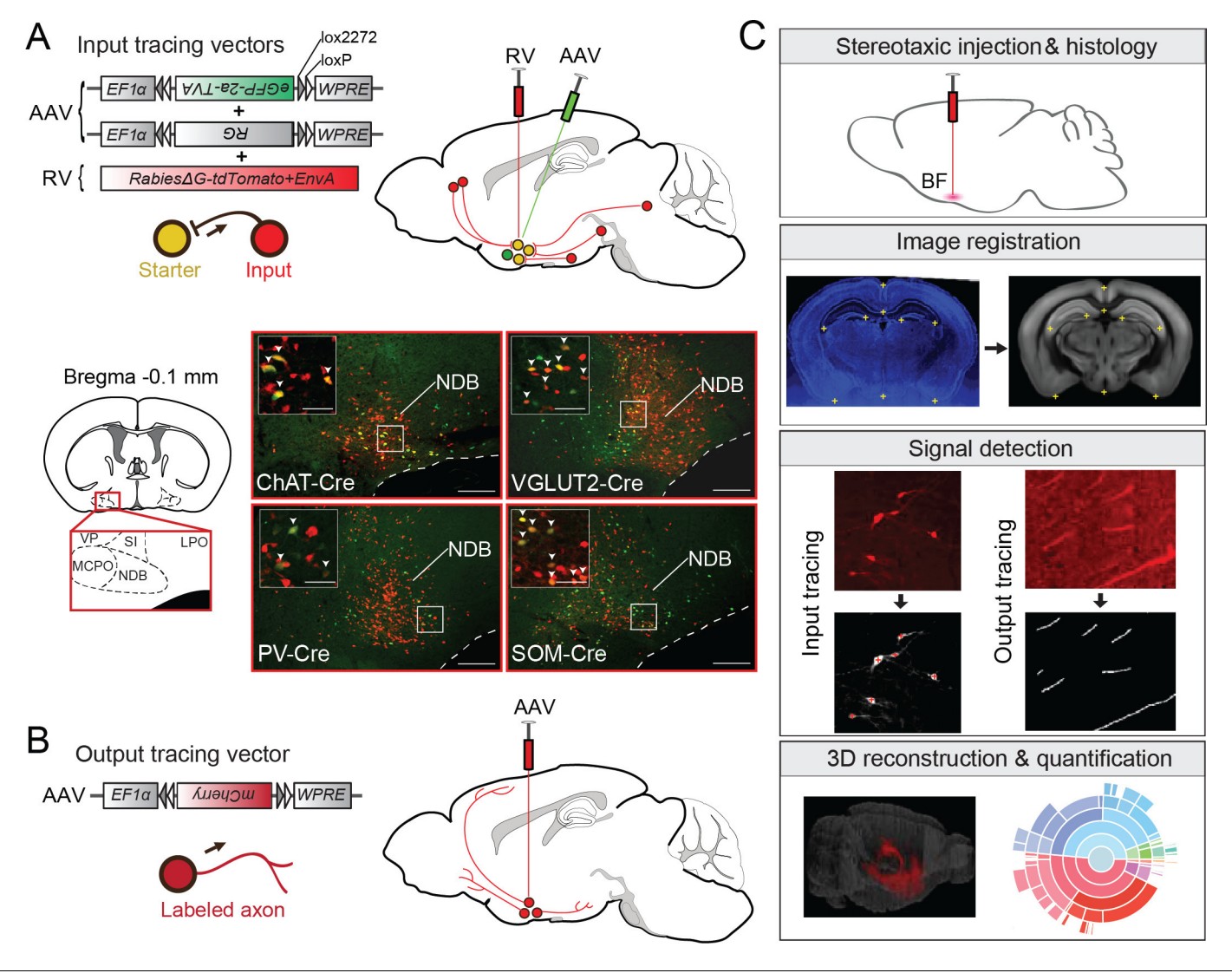

**Figure 1.** Experimental and analysis procedures for cell-type-specific circuit tracing. (**A**) RV-mediated transsynaptic retrograde tracing of BF inputs. Upper panel, viral vectors and injection procedure. Lower panel, fluorescence images of BF in the region of the NDB (red box in coronal diagram) in ChAT-, VGLUT2-, PV-, and SOM-Cre mice. Scale bar, 200 μm. Inset, enlarged view of the region in white box showing starter cells (yellow, expressing both eGFP and tdTomato, indicated by white arrowheads). Scale bar, 50 μm. NDB, diagonal band nucleus; SIB, substantia innominata, basal part; MCPO, magnocellular preoptic nucleus; VP, ventral pallidum; LPO, lateral preoptic area. (**B**) Viral vector and injection procedure for tracing BF axonal projections. (**C**) Flow chart showing the main steps in data generation and processing.

Lower panel, brain outline adapted from Figure 32 from The Mouse Brain in Stereotaxic Coordinates, 3rd edition, Franklin, K.B.J. and Paxinos, G (© copyright Elsevier, 2008. All Rights Reserved).

The following figure supplements are available for figure 1:

**Figure supplement 1.** Cell-type specificity of Cre-dependent rabies glycoprotein expression.

**Figure supplement 2.** Control experiments for RV tracing of inputs.

**Figure supplement 3.** Heat map distribution of starter cells.

**Figure supplement 4.** The relationship between the numbers of starter cells and input cells.

neurons were counted only in coronal sections outside of this range. While this procedure precludes identification of local inputs, synaptic interactions of the four cell types within the BF have been characterized electrophysiologically in a recent study (*Xu et al., 2015*). Another technical limitation of the study is that when the brain was removed for histological processing, the olfactory bulb was often damaged, which led to a significant underestimation of labeling (both the input neurons and axon projections) in the olfactory bulb.

We found 900 – 14,631 (median 7002) tdTomato-labeled presynaptic neurons in each brain (n = 17), and the convergence index (ratio between the number of input cells and starter cells) ranged between 4.3 and 77.7 (*Figure 1—figure supplement 4*). Such variability is comparable to that found in other studies using similar methods (*Miyamichi et al., 2011*; *DeNardo et al., 2015*). The presynaptic neurons were predominantly ipsilateral to the starter population (<5% contralateral) but were distributed in a large number of brain areas (*Figure 2*, *Figure 3*, *Video 1*). Since the number of labeled neurons varied across brain samples, and there was no significant difference between the four cell types (*P* = 0.27, one-way ANOVA), we normalized the data in each area by the total number of labeled neurons in each brain. When the brain was divided into 12 major regions (*Figure 3A*), the striatum and hypothalamus provided the highest numbers of inputs, while few labeled neurons were found in the medulla or cerebellum (*Figure 3A*).

To facilitate data visualization at different levels of detail, we also used an interactive sunburst diagram (adapted from Allen Mouse Brain Atlas, http://www.brain-map.org/api/examples/examples/sunburst/) to represent the whole-brain distribution of inputs to each cell type (http://sleepcircuits.org/bf/). The brain structures are arranged hierarchically from inner to outer circles, and the size of each sector represents the percentage of input from the corresponding structure. The name of each structure and its input percentage can be read out by pointing the cursor, and each region of interest can be expanded with a mouse click.

When the input distribution was analyzed at a finer spatial scale (e.g., the 6th ring of the sunburst plot), the nucleus accumbens (*Mesulam and Mufson, 1984*; *Zaborszky and Cullinan, 1992*), lateral hypothalamus (*Cullinan and Zaborszky, 1991*; *Grove, 1988b*; *Mesulam and Mufson, 1984*), and central nucleus of the amygdala (*Grove, 1988b*; *Pare and Smith, 1994*) were among the structures containing the highest numbers of input neurons (*Figure 2*). Interestingly, many close neighbors of these densely labeled structures (e.g., the basolateral nucleus of the amygdala, immediately adjacent to the central nucleus) showed very sparse or no labeling, indicating high spatial specificity of the long-range inputs. On the other hand, the input distributions were qualitatively similar between cell types, although with quantitative differences. For example, glutamatergic neurons received significantly more inputs from the lateral hypothalamus than the other cell types (P = 0.001, VGLUT2 vs. ChAT; P = 0.001, VGLUT2 vs. PV; P = 0.001, VGLUT2 vs. SOM;, one-way ANOVA and post-hoc Tukey's test), and PV+ neurons received more inputs from the nucleus accumbens (ACB) (P = 0.004, PV vs. ChAT; P = 0.003, PV vs. VGLUT2; P = 0.001, PV vs. SOM; one-way ANOVA and post-hoc Tukey's test).

To further verify the inputs revealed by RV-mediated retrograde tracing, we optogenetically tested the synaptic connections from the prefrontal cortex (PFC) and ACB (*Figure 4*). To verify the innervation from PFC to BF cholinergic neurons, we injected AAV (AAV-DJ-CaMKIIα-hChR2-eYFP) expressing the mammalian codon-optimized channelrhodopsin-2 (hChR2) fused with enhanced yellow fluorescent protein (eYFP) in the orbital and agranular insular areas of the PFC (*Figure 4—figure supplement 1*) in ChAT-eGFP mice and made whole-cell voltage-clamp recordings from eGFP-labeled cholinergic neurons in acute BF slices (*Figure 4A*). Activating the ChR2-expressing axon terminals with blue light evoked excitatory responses in all recorded BF cholinergic neurons (*n* = 9, *Figure 4B and C*), confirming the input revealed with RV tracing. To confirm the innervation from ACB, we injected Cre-inducible AAV (AAV-DJ-EF1α-FLEX-ChR2-eYFP) expressing ChR2-eYFP in ACB of GAD2-Cre mice, made whole-cell current-clamp recordings from unlabeled postsynaptic BF neurons, and used single cell reverse-transcription PCR (RT-PCR) to identify the cell type. We found that all four BF cell types received inhibitory responses from the ACB (*Figure 4D and E*; ChAT+: 2 out 5 showed significant responses; VGLUT2+: 2/4; PV+: 3/3; SOM+: 4/8), which is consistent with the finding of an electron microscopic double-immunolabeling study performed in rats (*Zaborszky and Cullinan, 1992*).

We next mapped the output of each BF cell type. To label the axonal projections, we injected AAV with Cre-dependent expression of mCherry (*Figure 1B*) into the BF of ChAT-, VGLUT2-, PV- or SOM-Cre mice. Two to three weeks after injection, the brain tissues were processed, images were

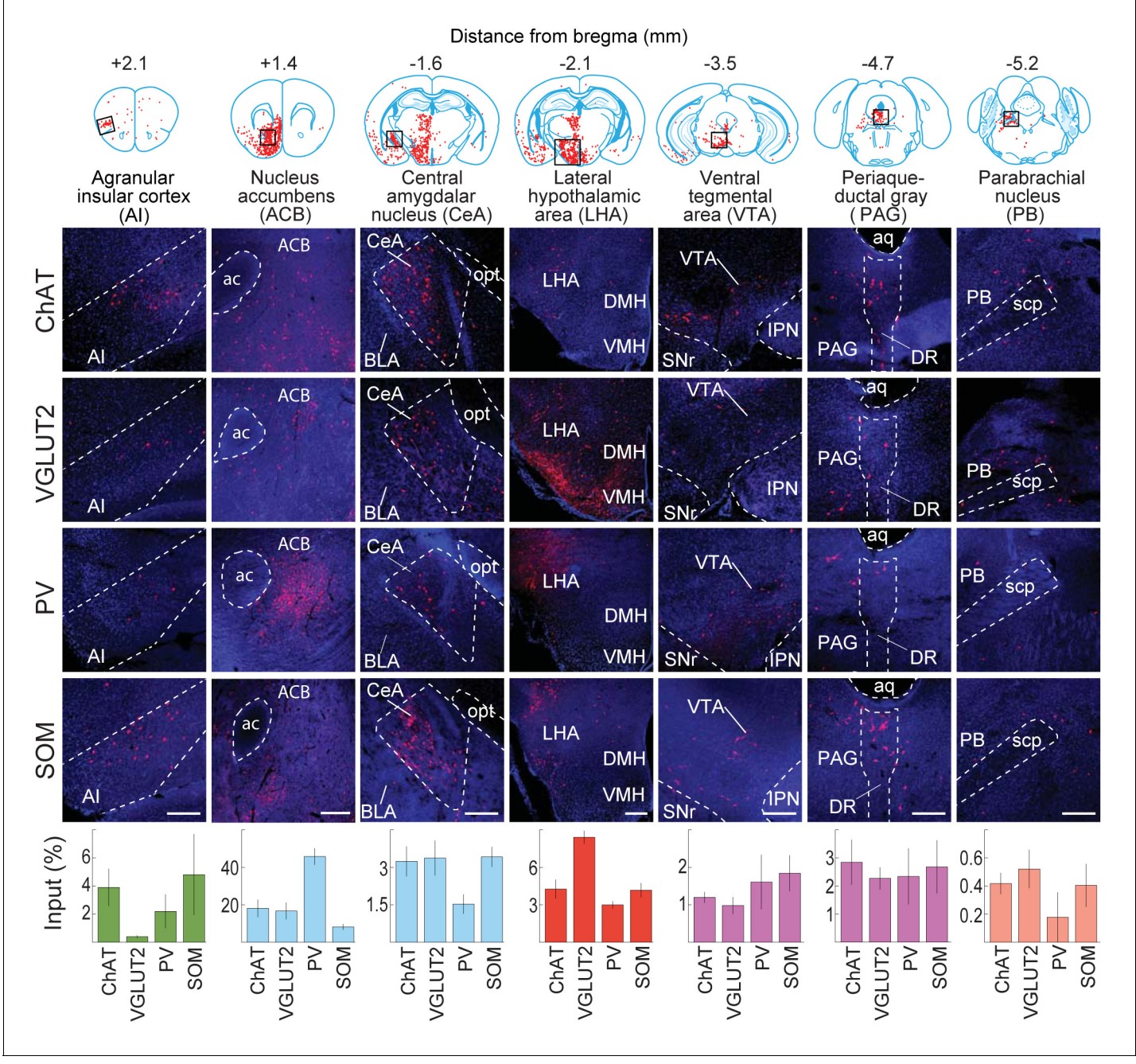

**Figure 2.** Inputs to each BF cell type from selected brain regions. Examples of RV-labeled input neurons to each of the four BF cell types in seven selected brain structures (black box in each coronal diagram). Scale bar, 200 μm. In each coronal diagram, RV-labeled neurons detected in all four brain samples are indicated by red dots. Bottom panel, mean percentage of input neurons in each brain structure for the four BF cell types. Error bar, ± s. e.m. Bar color indicates which of the 12 regions the given brain structure belongs to as depicted in *Figure 3*. ac, anterior commissure; aq, cerebral aqueduct; BLA, basolateral amygdalar nucleus; DMH, dorsomedial nucleus of the hypothalamus; DR, dorsal nucleus raphe; IPN, interpeduncular nucleus; opt, optic tract; scp, superior cerebellar peduncles; SNr, substantia nigra reticularis; VMH, ventromedial hypothalamic nucleus.
Upper panel, brain outlines adapted from Figures 13, 19, 44, 48, 60, 70, 74, from The Mouse Brain in Stereotaxic Coordinates, 3rd edition, Franklin, K.B. J. and Paxinos, G (© copyright Elsevier, 2008. All Rights Reserved).

registered to the reference atlas, and labeled axons were detected (*Figure 1C*, see Materials and methods). After the injection site (identified by the existence of labeled cell bodies) and locations

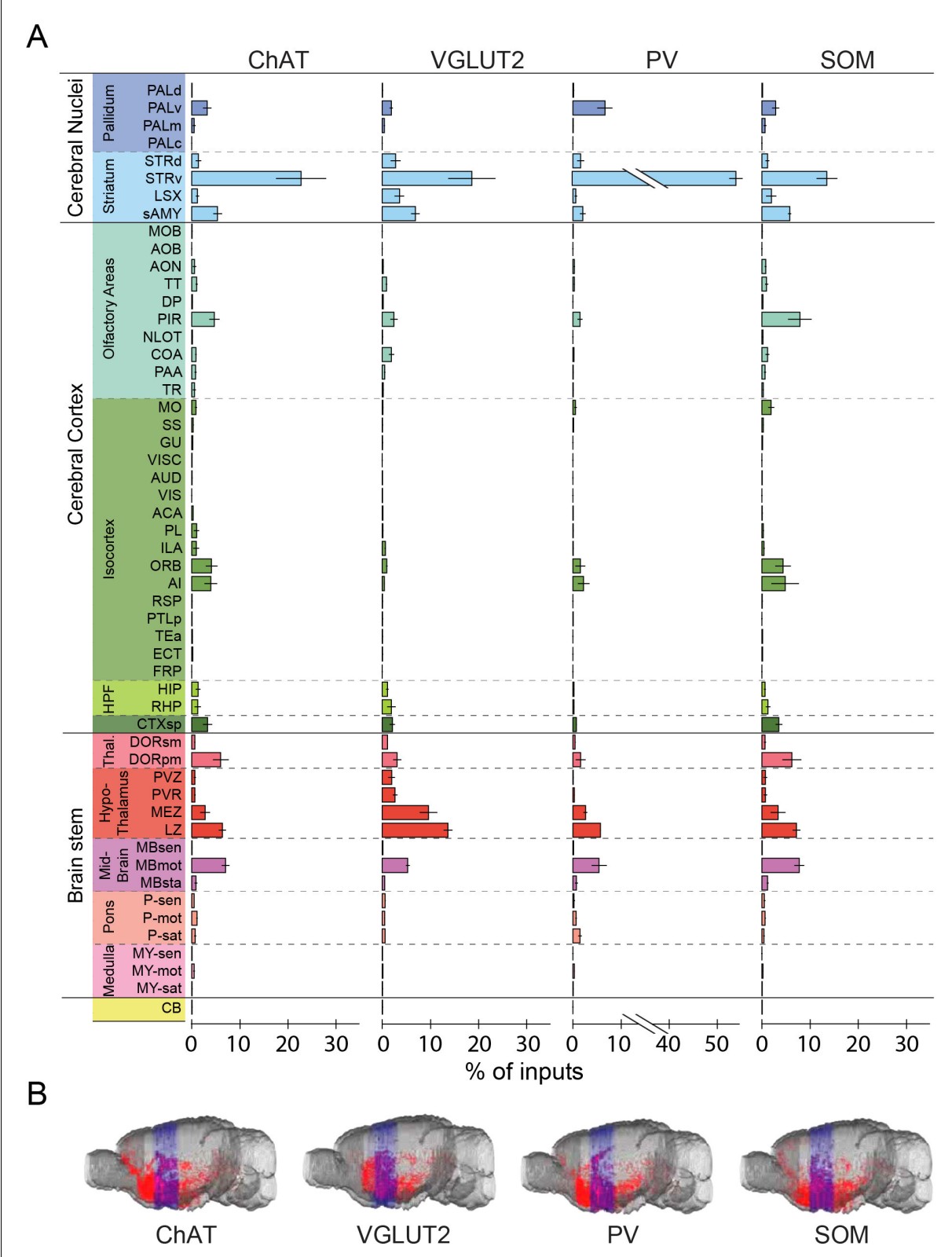

**Figure 3.** Whole-brain distributions of inputs to the four BF cell types. (**A**) Percentages of retrogradely labeled input neurons in 53 brain areas (ChAT, n = 5 mice; VGLUT2, n = 5; PV, n = 3; SOM, n = 4). Brain areas are grouped into 12 generalized, color-coded brain structures. HPF, hippocampal formation. Abbreviations of the 53 brain areas and their percentages of inputs are listed in *Figure 3—source data 1*. Error bar, ± s.e.m. Since labeled neurons in coronal sections near the injection site were excluded from analysis (see *Figure 1—figure supplement 2*), inputs from the pallidum are

*Figure 3 continued*

likely to be underestimated. (**B**) Whole-brain 3D reconstruction of the inputs to the four BF cell types. The blue-shaded area denotes the region excluded for analysis due to potential local contamination (see *Figure 1—figure supplement 2*).

The following source data is available for figure 3:

**Source data 1.** Distribution of input cells in 53 brain areas for ChAT+, VGLUT2+, PV+, and SOM+ BF neurons.

with known major fiber tracks were excluded, the projection to each brain area was quantified by the number of pixels occupied by the detected axons (*Oh et al., 2014*) (see Materials and methods).

Parallel to the broad distribution of inputs (*Figure 3*), we found that each BF cell type also projected to a large number of brain areas (*Figure 5*, *Figure 6*, *Video 2*, http://sleepcircuits.org/bf/, >95% ipsilateral). Among the 12 major brain subdivisions (*Figure 6A*), the hypothalamus, pallidum, and striatum received the heaviest BF projections (*Grove, 1988a*; *Henny and Jones, 2006*), while very few axons were detected in the medulla or cerebellum (*Figure 6A*). Analysis at finer scales revealed high spatial specificity of the projections. For example, while several cell types projected strongly to the lateral habenula (*Figure 5*), few axons were detected in the immediately adjacent but anatomically distinct medial habenula (*Hikosaka, 2010*). In addition to providing extensive inputs to the BF (*Figure 2*), the lateral hypothalamus was also a major recipient of BF projections (*Figure 5*), indicating a strong BF-hypothalamus loop that may be important for brain-state regulation (*Brown et al., 2012*; *Jones, 2011*; *Saper et al., 2010*). Importantly, whereas the input distributions were generally similar across BF cell types (*Figure 2*, *Figure 3*), the output patterns showed striking differences. For example, compared to the other cell types, the projection from cholinergic neurons was much stronger in the basolateral amygdala, hippocampus, and visual cortex but much weaker in the lateral hypothalamus, lateral habenula, and the ventral tegmental area (*Figure 5*). The different projection patterns among cell types are also apparent in the 3D whole-brain view (*Figure 6B*, *Figure 6—source data 1*).

To further compare the inputs and outputs between cell types, we averaged the spatial distributions across brain samples of each cell type and computed the correlation coefficient (CC) between cell types. For input distribution, the CCs between all cell types were high (*Figure 7A*), confirming their overall similarity observed earlier (*Figure 2*, *Figure 3*). On the other hand, when we computed the CCs between individual brain samples, we found higher CCs between samples of the same cell type (0.81 ± 0.04, s.e.m.) than of different cell types (0.70 ± 0.02, *P* = 0.01, *t*-test; *Figure 7—figure supplement 1A*). This indicates that despite

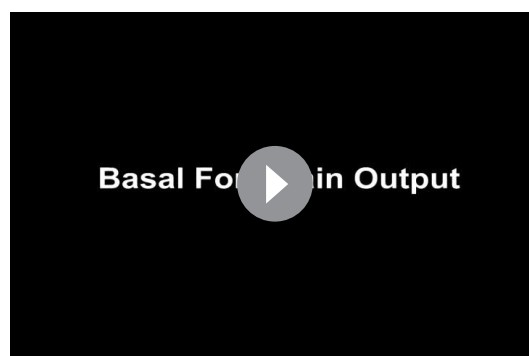

**Video 1.** 3D whole-brain view of RV-labeled inputs to ChAT+, VGLUT2+, PV+ and SOM+ BF neurons. Shown are data from four example brains (one for each cell type). Each red dot represents one RV-labeled presynaptic neuron. Blue, coronal sections within 850 µm from the injection site; neurons within this region were excluded from analyses to minimize contamination by the local background (*Figure 1—figure supplement 2*).

**Video 2.** 3D whole-brain view of mCherry-labeled axonal projections from ChAT+, VGLUT2+, PV+ and SOM+ BF neurons. Shown are projections averaged across all brain samples of each cell type.

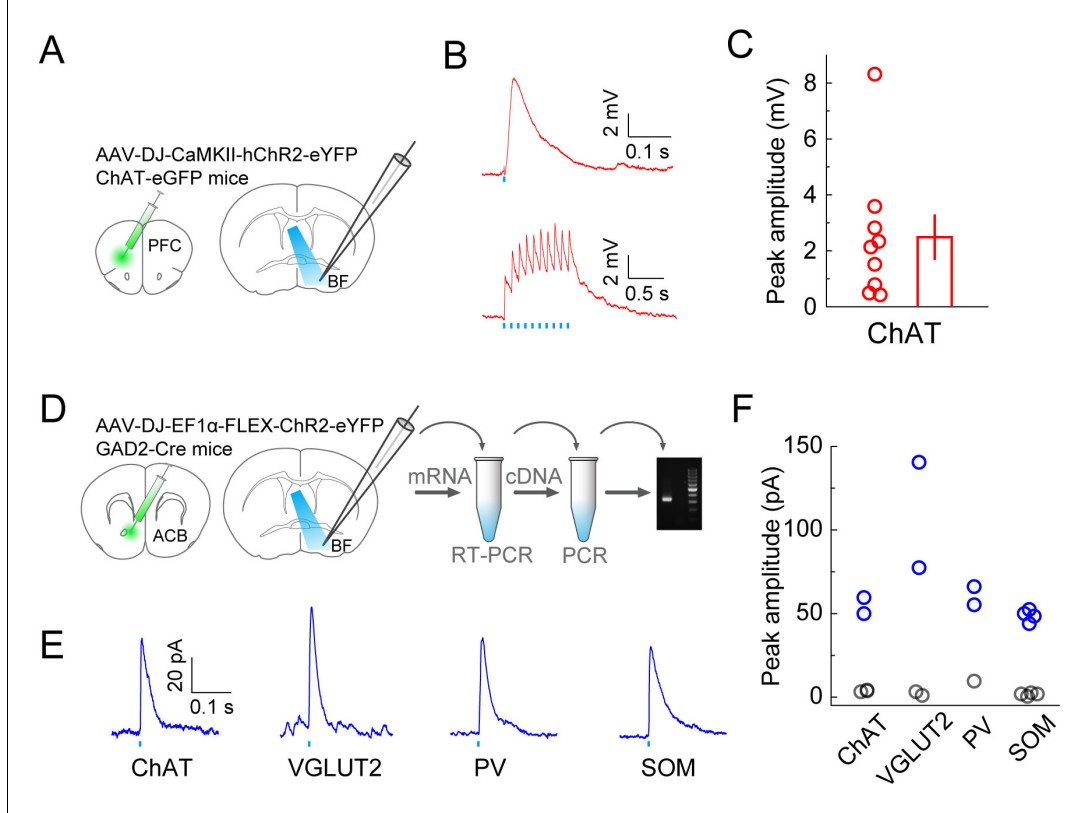

**Figure 4.** Optogenetic characterization of monosynaptic inputs to the BF from PFC and ACB. (A) Schematic of experiment. ChR2 was expressed in excitatory neurons in the prefrontal cortex of ChAT-eGFP mice by injecting AAV-DJ-CaMKIIα-hChR2-eYFP. Coronal slices of the BF were used for recording experiments. (B) Excitatory postsynaptic potentials recorded from ChAT+ neurons (under whole-cell current clamp) evoked by blue-light activation of the prefrontal cortical axons. Upper, response to a single light pulse (5 ms) in an example ChAT+ neuron; lower, responses to 10 pulses at 10 Hz recorded from a different ChAT+ neuron. (C) Summary of the peak amplitude of the response to a single light pulse. Each circle represents data from one BF ChAT+ neuron (n = 9 neurons from 2 mice). Bar, mean ± s.e.m. (D) Diagram illustrates virus injection site in the ACB and recording site in the BF. AAV-DJ-EF1α-FLEX-ChR2-eYFP was injected into the ACB of GAD2-Cre mice and whole-cell voltage-clamp recordings (clamped at 0 volts) were made from BF neurons. Single-cell gene-expression analysis was performed after each recording session to identify the cell type of each recorded neuron. (E) Example traces of laser-evoked responses in the four BF cell types. (F) Summary of the peak current amplitude of each neuron's response (ChAT+, n = 5 neurons from 5 mice; VGLUT2+, n = 4 neurons from 4 mice; PV+, n = 3 neurons from 3 mice; SOM+, n = 8 neurons from 4 mice). Gray indicates no significant response.

Panels A and D, brain outlines adapted from Figures 14, 18, 30, from The Mouse Brain in Stereotaxic Coordinates, 3rd edition, Franklin, K.B.J. and Paxinos, G (© copyright Elsevier, 2008. All Rights Reserved).

The following figure supplement is available for figure 4:

**Figure supplement 1.** Basal forebrain input from the prefrontal cortex.

the overall similarity, there were genuine differences between cell types that were beyond experimental variability.

For output distribution, the CCs between individual samples of the same cell type were also high (0.86 ± 0.05; *Figure 7—figure supplement 1B*), indicating reproducibility of the mapping. However, most of the CCs between cell types (*Figure 7B*, computed after averaging across samples of the same cell type) were much lower than those for input distribution. The two lowest CCs (ChAT+ vs. VGLUT2+ and PV+ neurons) reflect the fact that while the cholinergic neurons project strongly to structures within the cerebral cortex (including olfactory areas, isocortex, hippocampus, and cortical subplate) and weakly to the brain stem structures (thalamus, hypothalamus, and midbrain), glutamatergic and PV+ neurons (with output distributions highly similar to each other) showed complementary projection patterns (*Figure 6*).

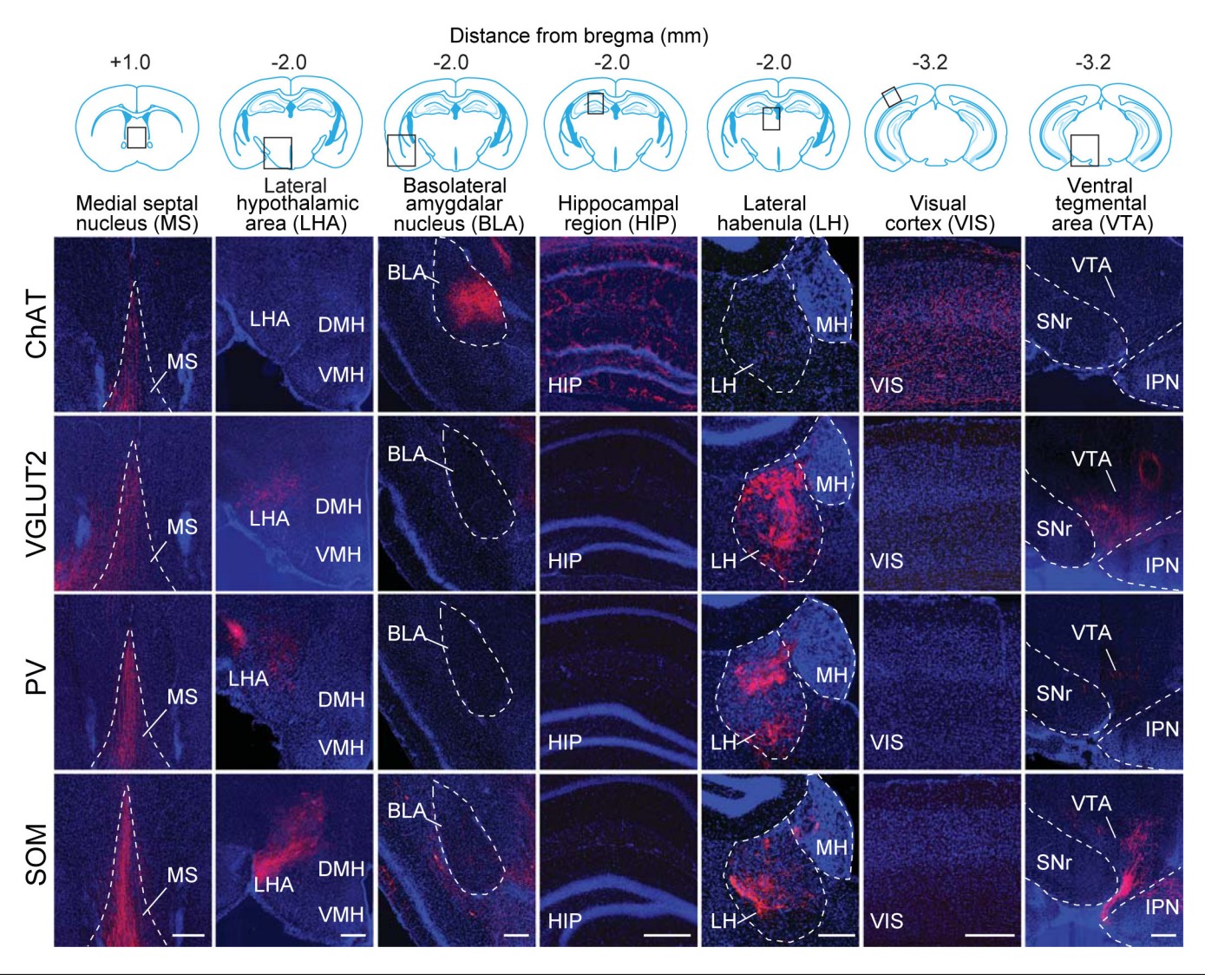

**Figure 5.** Axon projections of each BF cell type to selected brain regions. Examples of axon projections from each of the four BF cell types to seven selected brain structures (black box in each coronal diagram). Scale bar, 250 μm. DMH, dorsomedial nucleus of the hypothalamus; IPN, interpeduncular nucleus; MH, medial habenula; SNr, substantia nigra reticularis; VMH, ventromedial hypothalamic nucleus.
Upper panel, brain outlines adapted from Figures 23, 48, 57, from The Mouse Brain in Stereotaxic Coordinates, 3rd edition, Franklin, K.B.J. and Paxinos, G (© copyright Elsevier, 2008. All Rights Reserved).

Finally, we computed the CC between the input and output distributions of each cell type (*Figure 7C*). The highest CC was found for SOM+ neurons, reflecting their strong reciprocal connections with a number of brain structures, including the hypothalamus, striatum, pallidum, and olfactory areas (*Figure 7D*, lower right, *Figure 7—source data 1*. For glutamatergic neurons, the high CC reflects their strong reciprocal connections with the hypothalamus and striatum (*Figure 7D*, upper right). For cholinergic and PV+ GABAergic neurons, the CCs between input and output distributions were much lower, reflecting the facts that while both cell types receive strong input from the striatum, cholinergic neurons project strongly to the cerebral cortex, and PV+ neurons to the pallidum and hypothalamus (*Figure 7D*, upper and lower left).

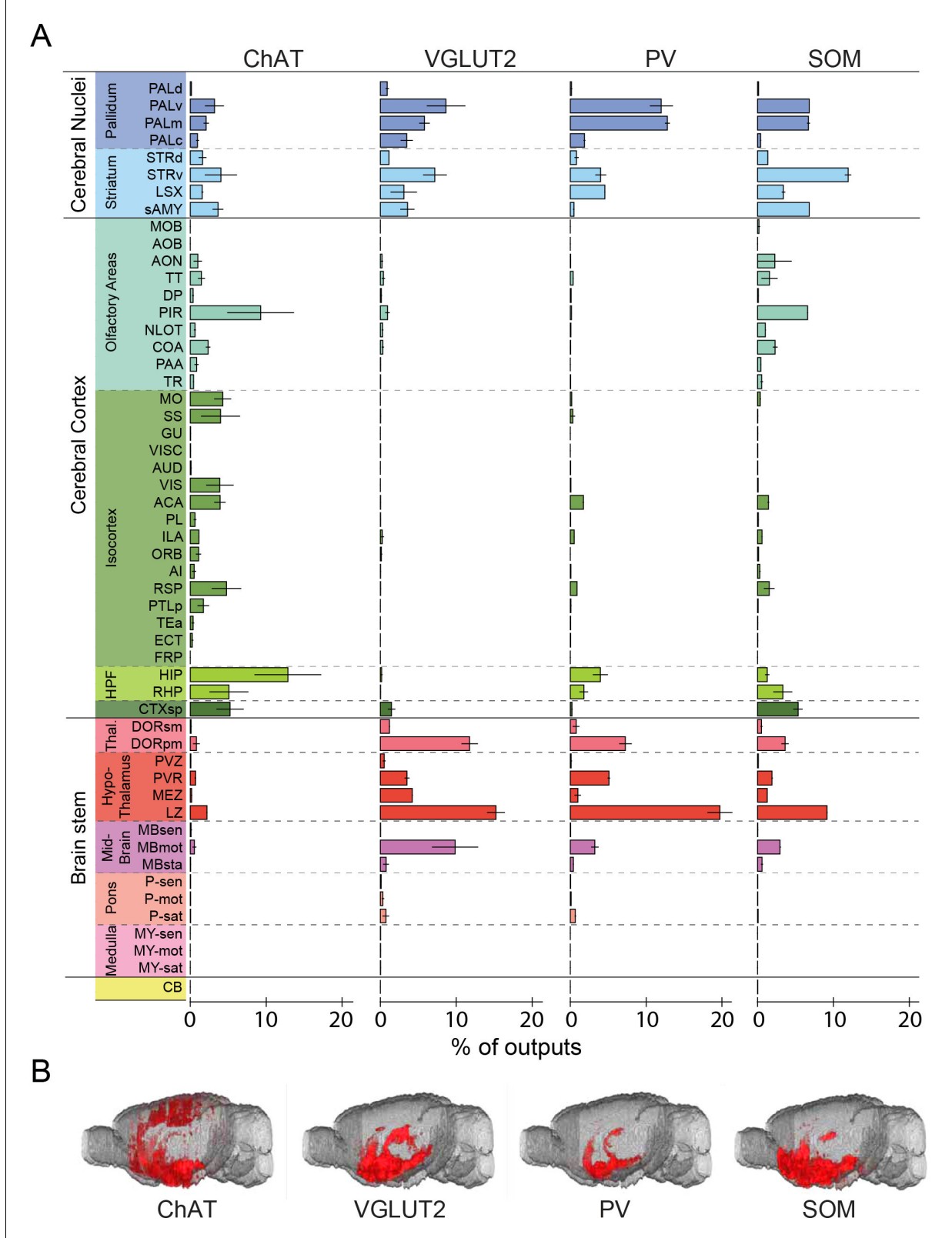

**Figure 6.** Whole-brain distributions of axonal projections from the four BF cell types. (A) Percentages of labeled axons in 53 brain areas (ChAT, n = 3 mice; VGLUT2, n = 3; PV, n = 3; SOM, n = 3). Error bar, ± s.e.m. Abbreviations of the 53 brain areas and their percentages of inputs are listed in *Figure 6—source data 1*. (B) Whole-brain 3D reconstruction of axon projections from each of the four BF cell types. Note that although VGLUT2+ and PV+ neuron projections showed the similar spatial distribution, there were fewer labeled axons from PV+ than VGLUT2+ neurons.
*Figure 6 continued on next page*

*Figure 6 continued*

The following source data is available for figure 6:

**Source data 1.** Distribution of axonal projections to 53 brain areas from ChAT+, VGLUT2+, PV+, and SOM+ BF neurons.

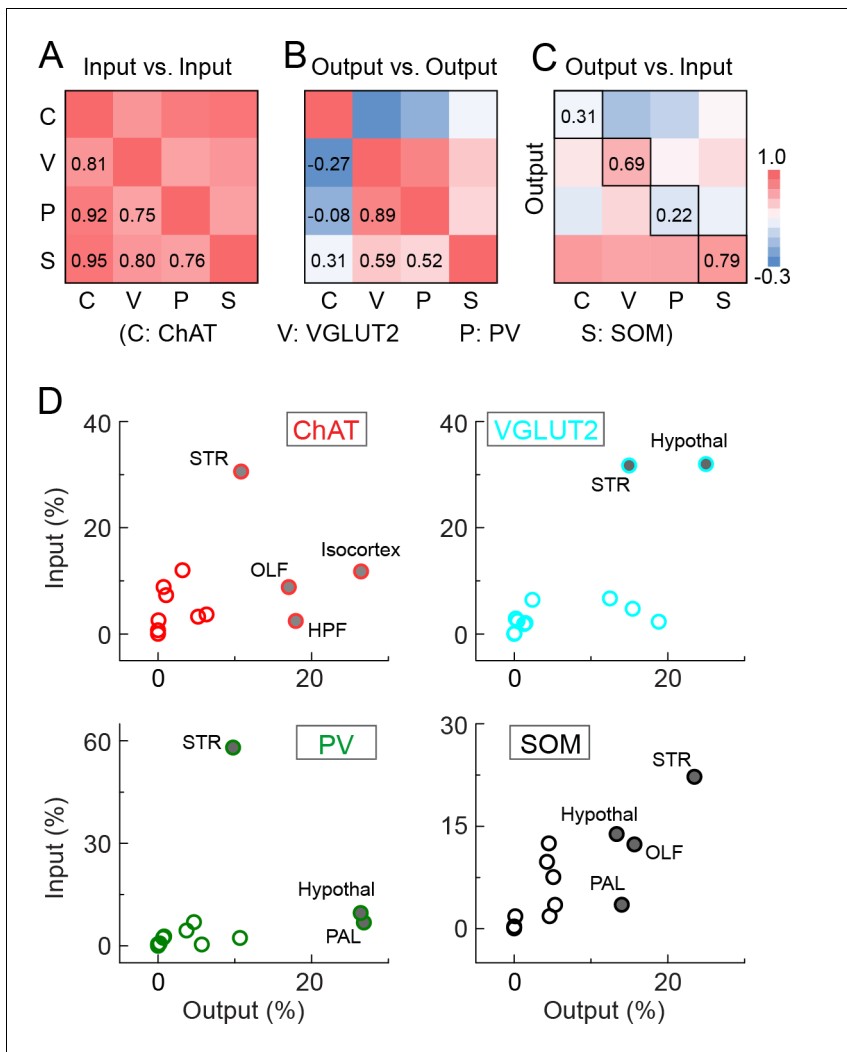

**Figure 7.** Comparison of input and output distributions. (**A**) Matrix of correlation coefficients (CCs) between input distributions of each pair of cell types. (**B**) Similar to **A**, for output distributions. (**C**) CCs between input and output distributions. All CCs were computed at the spatial scale of the 12 major brain subdivisions (*Figure 7—source data 1*). (**D**) Percentage of input vs. percentage of output in each region, for each of the four BF cell types. Filled circles, strongly connected brain regions contributing to the high CCs for glutamatergic and SOM+ neurons and low CCs for cholinergic and PV+ neurons in **C**.

The following source data and figure supplement are available for figure 7:

**Source data 1.** Distribution of BF input and output from 12 major brain subdivisions across cell-type.

**Figure supplement 1.** Correlation coefficients between individual brain samples for input and output distributions.

## Discussion

Using virus-mediated circuit mapping, we have characterized the whole-brain distributions of BF long-range connections, available in an open-access online database (http://sleepcircuits.org/bf/). Our experiments confirmed many previously demonstrated connections, but with cell-type specificity and quantitative analyses at multiple spatial scales. For example, we found that cortical inputs (*Mesulam and Mufson, 1984*; *Steriade and McCarley, 2005*) to all four BF cell types originate primarily from the agranular insular and orbital areas of the prefrontal cortex (*Figure 2*). While a previous ultrastructural study failed to detect convincing synaptic contact between prefrontal axons and BF cholinergic neurons (*Zaborszky et al., 1997*), our RV-mediated transsynaptic tracing demonstrated extensive monosynaptic innervation, which was also validated by electrophysiological recordings (*Figure 4B and C*). These findings have important implications on how the prefrontal cortex may exert top-down control of neural processing through its projection to the BF (*Sarter et al., 2001*). A recent study showed that cholinergic neurons in the BF are strongly activated by reinforcement signals during an auditory detection task (*Hangya et al., 2015*). Our whole-brain mapping of their inputs provides a list of candidate neurons through which the reinforcement signals are conveyed to the BF cholinergic neurons.

Regarding the outputs, we found striking differences across cell types (*Figure 6*). A recent study has shown that cholinergic, glutamatergic, and PV+ neurons all promote wakefulness, while SOM+ neurons promote sleep (*Xu et al., 2015*). The distinct projection patterns between cholinergic and glutamatergic/PV+ neurons (*Figure 6*) suggest that they preferentially regulate different brain functions during wakeful states. In a recent study, optogenetic activation of BF PV+ neurons was shown to enhance cortical gamma band oscillations (*Kim et al., 2015*). In addition to direct projections to the cortex, our study showed extensive subcortical projections of PV neurons, which may also contribute to the regulation of cortical gamma oscillations. The output distribution of SOM+ neurons, on the other hand, was highly correlated with the input distributions of all BF cell types (*Figure 7C*, bottom row); the broad GABAergic inhibition of these input areas by SOM+ neurons may be important for the sleep-promoting effect. Thus, while the highly convergent inputs from multiple brain areas allow a variety of sensory, motor, cognitive, and emotional signals to be integrated within the BF, the distinct projections by different cell types may enhance the versatility of the BF in coordinating diverse functions of multiple brain networks.

## Materials and methods

### Virus preparation

#### Transsynaptic retrograde tracing

To construct AAV2-EF1α-FLEX-eGFP-2a-TVA and AAV2-EF1α-FLEX-RG, TVA and eGFP linked by the 2A 'self-cleaving' peptide or rabies glycoprotein was respectively cloned into pAAV-MCS (Stratagene, La Jolla, CA) in an antisense direction flanked by a pair of canonical loxP sites and a pair of lox2272 sites. AAV particles (AAV2/2) were produced by co-transfection of packaging plasmids into HEK293T cells, and cell lysates were fractionated by iodixanol gradient ultracentrifugation. Viral particles were further purified from the crude fraction by heparin affinity column (HiTrap Heparin HP Columns; GE Healthcare, Pittsburgh, PA), desalted and concentrated with Amicon Ultra Centrifugal Filter (100 K, Millipore, Bellerica, MA). The genomic titer of AAV2-EF1α-FLEX-eGFP-2a-TVA (4.4 × $10^{13}$ gc/ml) and AAV2-EF1α-FLEX-RG (2.2 × $10^{12}$ gc/ml) was estimated by quantitative PCR. eGFP-2a-TVA and rabies glycoprotein were subcloned from the AAV-TRE-HTG plasmid from L. Luo.

RV-ΔG-tdTomato was amplified in B7GG cells and pseudotyped using BHK-EnvA cells. EnvA pseudotyped rabies virus was titered (1.5 × $10^9$ IU/ml) by infecting the 293T-TVA8000 (*Narayan et al., 2003*) cell line with serial dilutions of the stock virus. RV-ΔG-tdTomato was a gift from B. Lim. B7GG cells, BHK-EnvA cells (*Wickersham et al., 2007*), and 293T-TVA8000 cells were gifts from E. Callaway.

#### Anterograde axon tracing

AAV2-EF1α-FLEX-mCherry was purchased from the UNC Vector Core (Chapel Hill, NC) and the titer was estimated to be ~$10^{12}$ gc/ml.

## Surgery and viral injections

All experimental procedures were approved by the Animal Care and Use Committee at the University of California, Berkeley. For the current study, we targeted the caudal portion of the BF (including the horizontal limb of the diagonal band of Broca, magnocellular preoptic nucleus, and substantia innominata) rather than the rostral nuclei (medial septum and the vertical limb of the diagonal band of Broca). For virus injection, adult (>P40) $Chat^{tm2(cre)Lowl}$ (ChAT-Cre, JAX#006410), $Slc17a6^{tm2(cre)Lowl}$ (Vglut2-Cre, JAX#016963), $Pvalb^{tm1(cre)Arbr}$ (PV-Cre, JAX#008069), and $Sst^{tm2.1(cre)Zjh}$ (SOM-Cre, JAX#013044) mice were anesthetized with ~1.5% isoflurane in oxygen (flow rate of 1L/min). A craniotomy (~0.5 mm diameter) was made at 0.1 mm posterior to bregma, 1.3 mm lateral to midline. For anterograde axon tracing, 300–400 nL of AAV (serotype 2) expressing Cre-dependent mCherry (AAV2-EF1α-FLEX-mCherry) was stereotactically injected into the BF (5.2 mm from brain surface) using Nanoject II (Drummond Scientific, Broomall, PA) via a micro pipette. The following steps were taken to minimize virus leaking into the injection track: (1) The pipette opening was minimized (<20 µm); (2) The injector was mounted onto a motorized manipulator to ensure slow and smooth retraction; (3) The injection started 5 min after pipette insertion, and multiple 23 or 40 nl injections (13 nl/s) were made at 15–30 s intervals. The pipette was retracted 10 min after injection.

For transsynaptic retrograde tracing, 200–300 nl of helper AAV (AAV2-EF1α-FLEX-eGFP-2a-TVA and AAV2-EF1α-FLEX-RG mixed at 1:1 ratio of viral particles) was injected into the BF using the same procedure as described above. Two to three weeks after helper AAV injection, RVΔG-tdTomato+EnvA was injected into the same location. To further ensure localized virus expression, the helper AAV injection pipette was tilted at 20 degrees from vertical while RV injection pipette was inserted vertically in the majority of experiments.

## Tissue processing

Brain tissue was processed according to standard procedures. In brief, two to three weeks after AAV injection (for anterograde tracing) or one week after RV injection (for retrograde tracing), mice were deeply anesthetized with isoflurane and immediately perfused intracardially with ~15 ml of phosphate-buffered saline (PBS) (pH 7.2) followed by ~15 ml of 4% paraformaldehyde (PFA) in PBS. Brain tissue was carefully removed, post-fixed in 4% PFA in PBS at 4°C overnight, dehydrated in 30% sucrose in PBS for 48 hr, and embedded in Tissue Freezing Medium (Triangle Biomedical Sciences, Cincinnati, OH). Brains were cut in 30 or 50 µm coronal sections using a cryostat (Thermo Scientific, Waltham, MA) and mounted with VECTASHIELD mounting medium with DAPI (Vector Laboratories, Burlingame, CA) or DAPI Fluoromount-G (Southern Biotech, Birmingham, AL). One out of every three sections were imaged using 20X/0.75 objective in a high-throughput slide scanner (Nanozoomer-2.0RS, Hamamatsu, Japan) for further processing. We also imaged selected brain regions (*Figures 2* and *3*) using a Zeiss (Germany) inverted AxioObserver Z1 fully motorized microscope with LSM 710 confocal scanhead, 10X/0.3 EC Plan Neofluar M277 objective or a 20X/0.8 Plan Apochromat M27 objective.

## Immunostaining

To check for cell-type specific expression of rabies glycoprotein, tdTomato transgenic reporter mice (JAX#007914) were crossed to Cre-transgenic mice for each cell type and double transgenic offspring were injected with AAV2-EF1α-FLEX-RG. Alternatively, Cre-transgenic mice were injected with AAV2-EF1α-FLEX-mCherry and AAV2-EF1α-FLEX-RG.

After making coronal sections, brain slices were washed in PBS (3 x 10 min., room temperature), blocked with mouse IgG blocking reagent (Mouse on Mouse (M.O.M.) Kit, Vector Laboratories, Burlingame, CA) for 2 hr at room temperature, incubated with mouse anti-rabies glycoprotein (clone 24-3F-10, EMD Millipore, Billerica, MA) with M.O.M. protein concentrate in PBST (PBS + 0.3% Triton-X100) for 18 hr at room temperature, washed in PBST (3 x 20 min., room temperature), incubated with Alexa-Fluor 488 or 647 donkey anti-mouse (ThermoFisher Scientific, Waltham, MA) with M.O.M protein concentrate in PBST for 3 hr at room temperature, and finally washed in PBST (3 x 10 min, room temperature) prior to mounting the slides.

## 3D reconstruction and quantification

A software package was developed in Matlab to analyze the digitized brain images. The analysis software consists of three modules: image registration, signal detection, and quantification/visualization.

### Registration module

The registration module is a reference point-based image alignment software used to align images of brain sections to the Allen Mouse Brain Atlas for further quantification and 3D reconstruction. First, we manually selected a set of reference points in both the atlas and the brain image. The module then applied several geometric transformations (translation, rotation and scaling) of the brain section to optimize the match of the reference points between the brain image and the atlas. Since histological sectioning can sometimes cause tissue compression, we allowed the scaling factors along the dorsal-ventral and medial-lateral axes to be optimized independently. Following the transformation, the match between the image and the atlas was inspected, and further adjustments were made manually if necessary. The main purpose of the manual adjustment was to correct errors generated by the registration procedure due the imperfect brain slice preparations, and it was mostly performed by research assistants not involved in the research design and unaware of the final conclusion of the study.

### Detection module

The detection module has two independent sub-modules designed for counting RV-labeled cells and detecting axons, respectively. The cell counting module records the position of manually identified tdTomato-labeled neurons in each digitized brain sections image.

For axon detection, the ridge detection method was used (http://en.wikipedia.org/wiki/Ridge_detection). The following steps were taken to maximize the detection accuracy: (1) Image ridges were computed across multiple scales to extract all possible axon-like signals from each image. In the resulting binary 'ridge image', the number of pixels occupied by each detected axon depends on the length but not the thickness of the axon. In addition to valid axons, the ridge image also contains many noise pixels. (2) To remove the noise pixels due to the general background in the fluorescence image, we set a threshold based on the intensity distribution of the original image, and use this as a mask to remove the noise pixels in the ridge image obtained from step (1). (3) To remove the discrete noise pixels with fluorescence intensities higher than the general background (thus not removed by step 2), we first identified pixels that are spatially contiguous in the ridge image, computed the size of each contiguous region, and removed the regions below a threshold size. Steps 2 and 3 were repeated until satisfactory detection results were achieved. (4) The results were then visually inspected and the remaining noise pixels, which were mostly artifacts introduced during brain tissue processing, were rejected manually.

### Quantification/visualization module

After detection and registration, signals were quantified across the whole brain and projected to the 3D reference atlas for better visualization. The 3D viewer plug-in of the ImageJ software was used to animate the final 3D model.

The atlas, 3D reference mouse brain, quantification ontology, and layouts for sunburst plot were obtained from the open online resource of Allen Institute for Brain Science, licensed under the Apache License (Version 2.0). Since the number of labeled neurons or axons varied across brains, the input from each region was quantified by dividing the number of labeled neurons found in that region by the total number of labeled neurons detected >850 µm from the injection site (see *Figure 1—figure supplement 2*). The output (axon projection) to each region was quantified as the number of pixels occupied by detected axons in the cleaned ridge image (*Oh et al., 2014*) (see Detection module above) divided by the total number of axon-occupied pixels found in the entire brain (after excluding the injection site and locations with known major fiber tracks).

## Starter cell mapping

Starter cells were manually identified from the colocalization of tdTomato and eGFP signals using Nanozoomer images scanned at multiple focal planes. The starter cells were marked using the Cell

Counter ImageJ plug-in, and registered to the Allen Brain Reference Atlas as described above. A starter cell heat map was generated in Python by calculating the normalized starter cell density for all samples from each cell type and applying bicubic interpolation. For each coronal section image, the cell density was binned from an anterior-posterior range of 0.24 mm, centered at the listed brain slice coordinate (*Figure 1—figure supplement 3*).

## Slice recording

To validate the synaptic input from the prefrontal cortex to BF cholinergic neurons (as shown by RV-mediated input tracing), ChR2 was expressed in excitatory neurons in the prefrontal cortex of ChAT-eGFP mice (JAX#007902, P16-P18) by injecting ~500 nl of AAV-DJ-CaMKIIα-hChR2-eYFP (~$10^{13}$ gc/ml, Stanford Gene Vector and Virus Core, Stanford, CA) into the orbital and agranular insular areas of the PFC (2.0 mm anterior to bregma, 1.5 mm lateral, 2.0 mm from brain surface) and recording from eGFP+ BF neurons. To validate the synaptic input from ACB, Cre-inducible ChR2-eYFP was expressed in GAD2-Cre mice (JAX#010802, P16-P18) by injecting 300–400 nl of AAV-DJ-EF1α-FLEX-ChR2-eYFP (~$10^{13}$ gc/ml, Stanford Gene Vector and Virus Core, Stanford, CA) into the ACB (1.5 mm anterior to bregma, 0.8 mm lateral, 3.6 mm from brain surface) and recordings were made (one week after virus injection) from unlabeled BF neurons, which were identified after each recording via single-cell gene-expression analysis. Slice preparation, recording procedure, and single-cell gene-expression analysis were the same as described in a recent study (*Xu et al., 2015*).

## Acknowledgements

We thank K-S Chen, T Lei, D Jeong for the help with histological processing and data analysis, and B Lim and E Callaway for viral vectors and cell lines.

## Additional information

### Competing interests

LL: Reviewing editor, *eLife*. The other authors declare that no competing interests exist.

### Funding

No external funding was received for this work

### Author contributions

JPD, MX, S-HL, YD, Conception and design, Acquisition of data, Analysis and interpretation of data, Drafting or revising the article; W-CC, SZ, SC, TJY, JLF, Acquisition of data, Analysis and interpretation of data, Drafting or revising the article; KM, LL, Analysis and interpretation of data, Drafting or revising the article, Contributed unpublished essential data or reagents

### Author ORCIDs

Seung-Hee Lee, http://orcid.org/0000-0002-9486-5771
Tyler J Yung, http://orcid.org/0000-0002-8361-837X
Yang Dan, http://orcid.org/0000-0002-3818-877X

### Ethics

Animal experimentation: All surgical and experimental procedures were in accordance with the Care and Use of Laboratory Animals of the National Institutes of Health Guide and approved (#R229) by the Animal Care and Use Committees of University of California, Berkeley.

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
