## [Decision Letter]

[Editors’ note: this article was originally rejected after discussions between the reviewers, but the authors were invited to resubmit after an appeal against the decision.]

Thank you for submitting your work entitled "Cell Type-Specific Long-Range Connections of Basal Forebrain Circuit" for consideration by *eLife*. Your article has been reviewed by three peer reviewers, and the evaluation has been overseen by a Reviewing Editor and Gary Westbrook as the Senior Editor. Our decision has been reached after consultation between the reviewers. Based on these discussions and the individual reviews below, we regret to inform you that your work will not be considered further for publication in *eLife*.

The reviewers appreciated the importance of the work but raised important issues regarding the specificity of the viral approach used. The main issue raised, as seen in Figure 1—figure supplement 1, is that the virus(es) infected cells without Cre, i.e. they were leaky. Therefore, it is unclear how much of the mapping is specific. Besides quantification and better discussion of the extent of the leakage, the reviewers suggest that the study may be more suitable for publication if the mapping could be done with a more specific virus. Alternatively, or in addition, each proposed projection could be validated by expressing ChR2 in the presynaptic cell, and recording from the genetically defined postsynaptic cell. Either of these experiments would require a substantial amount of work in our opinion, and therefore, as per *eLife*'s policy, the study cannot be accepted.

Reviewer #1:

Do and colleagues perform a whole-brain mapping of both the inputs and outputs of the basal forebrain across its four major cell types using rabies virus. Understanding the connectivity of the basal forebrain is important and the new viral techniques used provide an unprecedented opportunity for revealing the cell-type-specific input-output logic of basal forebrain. These experiments required a lot of careful work and data analysis. The major take home message is that the inputs to all cell types are qualitatively similar but the output projections showed marked differences.

I was very enthusiastic about this paper until I got to supplementary Figure 1, the control experiment. What it shows is that the expression of rabiesdG-tdTomato+EnvA can occur without TVA or there is leaky expression of TVA without Cre (Figure 1). Figure 1 shows that about half of the rabiesdG-tdTomato+EnvA is not colocalized with AAV-TVA-GFP. These results indicate a severely leaky expression that causes non-specific infection of rabies in all cell types. The authors are to be commended for showing these controls. Clearly they realize their importance and try to focus on the relative fraction of inputs between cell-types. Unfortunately, however, having a mixed starter population can alone explain why the inputs of different cell types look the same, bringing the entire study into question.

Other comments:

1) The authors report that subsets of ChAT cells in the basal forebrain may project to or receive projections from different brain regions. Figure 1 showed the targeting injection site is a bit posterior and lateral compared to the usual coordinates of HDB, which may lead to only partial infection of HDB. In addition, the infected ChAT and PV neurons shown in Figure 1 are significantly fewer in number than the total number of ChAT and PV neurons in HDB. The author should stain for ChAT, PV, SOM, vGlut2 and quantify whether the infection includes at least most of the HDB neurons or just part of it.

2) A statement should be made with regard to whether the morphological analyses were made with the experimenter blinded to the treatment condition. This is especially important since all the quantification procedures from registration and detection module to cell counting are all manually performed or adjusted.

3) One particularly important piece of data in the paper is in Figure 2—figure supplement 1 showing physiological evidence for direct prefrontal inputs to cholinergic neurons. I think the authors should expand on these data and make into a full figure. There were not enough details provided to evaluate the experiment fully. The legend states the input is from "prefrontal cortex". Do the authors mean mPFC? Their rabies tracing data shows surprisingly little input from mPFC.

Reviewer #2:

In this study, the authors perform a detailed analysis of brain regions that input onto and receive outputs from four genetically defined cell-types in the basal forebrain. Using rabies-based labeling of monosynaptic inputs, they conclude that four major cell-types in the basal forebrain – PV+, SOM+, ChAT+, and VGLUT2^+^ neurons – receive qualitatively similar inputs, with striatum providing the main source of input. In addition, they report divergent outputs of each of these four cell-types as assessed by labeling of axons with an mCherry-expressing AAV, with ChAT+ neurons primarily projecting to the cortex, olfactory bulb, and hippocampus, PV+ neurons projecting to hypothalamus and pallidum, and both VGLUT2^+^ and SOM+ neurons forming reciprocal connections with the striatum and hypothalamus.

The analysis described in this study appear carefully done, and appropriate controls are shown. While the authors offer little in the way of interpretation, they have created a valuable, interactive resource that will be of great interest to other researchers interested in the basal forebrain.

There are several issues that should be addressed prior to publication. First, while the website is a valuable resource and welcome interactive supplement to the findings of the paper, there are some apparent discrepancies between the stated percentage values and the associated sunburst plot that either need to be addressed in the code or explained with better documentation or associated legends. For example, in the graphic showing inputs to ChAT cells, the "Basic cell groups and regions" lists 98.55% total inputs (it is understood from the legend of Figure 3—figure supplement 1 that some inputs come from unnamed sub-regions, though this could be mentioned in the main text). However, The 2nd level of the sunburst plot only includes "Brain stem" at 28.97% and "Cerebrum" at 64.80% – which sum to only 93.77%, not 98.44. As result, subsequent rings represent much more of the sunburst plot than the percentage would indicate – e.g. Inputs to ChAT neurons from Cerebral nuclei make up only 42.03% of the inputs, but well more than half of the sunburst plot. At a minimum this discrepancy needs to be better explained. Ideally, the sunburst plot would include an "un-named region" category, so that the graphic matches the percentage.

Perhaps the most striking finding of the paper is the major input to all cell-types from the striatum. To my knowledge, a major input from the striatum to the basal forebrain has not been previously described. Could this be the result of hitting starter cells in the pallidum during the rabies-assisted retrograde tracing? Confirming functional connectivity between striatal projection neurons and basal forebrain neurons of all types, as in Figure 2—figure supplement 1) would be very interesting and provide a necessary control to show that neurons of the BF proper also receive striatal input. This point needs to be addressed with electrophysiology and complete reconstruction as it would represent a radical change in our understanding of the BG wiring.

There is a great degree of variability in the number of labeled presynaptic neurons in different brains. Does this correlate in any way with the number of starter cells in each mouse? Greater analysis of the number and location of starter cells for each experiment would be useful.

Finally, the manuscript needs a greater discussion on the functional implications of the connectivity patterns discovered here, and the potential ways they do or do not meet expectations based on previous functional data. For example, the authors cite a recent paper showing PV+ BF neurons entrain cortical γ oscillations (Kim et al., 2015), but they show very little projection to the cortex from this population. The authors cite a recent paper (Hangya et al., 2015) that describes a recruitment of BF ChAT+ neurons by reinforcement with very long latency (~50 ms). Can the authors identify from their data an input that might explain this?

Reviewer #3:

Do et al. revealed input and output neural pathway to four different types of neurons (cholinergic, glutamatergic, parvalbumin (PV) and somatostain (SOM)) in the basal forebrain (BF) using Cre mice and transsynaptic retrograde tracing. They claimed that input pathway to four different cell types was similar but output pathway was different. Although the results were clear and informative to understand the physiological role of BF neurons, the following points need to be revised before publication.

Major points:

1) The author used cell type specific Cre recombinase-expressing mice (ChAT-Cre, VGLUT2-Cre, PV-Cre and SOM-Cre) to restrict starter cell for retrograde transsynaptic tracing. However, there is no information about specificity of Cre expression in each mouse line. How much Cre expressing neurons and which area of Cre expressing neurons were infected by rabies virus as starter cell should be provided.

2) Figure 2—figure supplement 1, the authors showed functional connection between prefrontal cortical (PFC) neurons and cholinergic neurons in BF. However, there is no information about feature of neurons in PFC labeled by retrograde transsynaptic tracing. In spite of this, the author used CaMKIIa promoter, which drives gene expression in glutamatergic neurons. Infected area of ChR2 expressing AAV in the PFC should be provided as well. Additionally, it is also important to show that brain area which is not innervate BF neurons from transsynaptic tracing is not functionally connected by expressing ChR2. This figure is referred only in the Discussion.

[Editors’ note: what now follows is the decision letter after the authors submitted for further consideration.]

Thank you for resubmitting your work entitled "Cell Type-Specific Long-Range Connections of Basal Forebrain Circuit" for further consideration at *eLife*. We apologize that this review took longer than is our goal. Your revised article has been favorably evaluated by Gary Westbrook (Senior editor), a Reviewing editor, and two reviewers, a reviewer from the original version of your manuscript and a new reviewer. The manuscript has been improved but there are some remaining issues that need to be addressed before acceptance, as outlined below.

Summary:

Overall this is nice study showing differential outputs, but largely similar inputs, from/to four different neurotransmitter-defined cell groups in the basal forebrain. The authors use optogenetics in slice recordings to confirm some of the connections identified using monosynaptic tracing with rabies virus, including a connection (from PFC to cholinergic BF neurons) that was not reported in a previous study using electron microscopy. The authors have done a good job of making the data available as a resource for subsequent investigations. The authors have made changes to the manuscript and it is definitely improved. The data showing direct cortical projections to all basal forebrain cell-types (Figure 4) is important and nice.

However there are still concerns about the controls for cell-type specificity that need to be addressed before a decision on publication can be made. Note that points 2 and 3 are overlapping and if the control experiment works it is one experiment that shows specific expression of RG along with the cell-type-specific markers as we already know TVA is leaky.

1) The control experiments described in Supplementary Figure 1 are good controls to do, and the ones in wild-type mice are fine (and important). For the controls in Cre mice, however, the authors use the wrong mouse line, for no apparent reason. The authors state that the use of GAD2-Cre mice "is likely to cause an overestimate of the exclusion zone"; this is plausible in the case of the PV-Cre and SOM-Cre lines, but not necessarily in the case of the Vglut2-Cre ones. It is known that EnvA-enveloped rabies virus can directly (non-transsynaptically) retrogradely infect TVA-expressing neurons projecting to an injection site (Huang & Hantman 2013). In the worst case in the present study, direct retrograde infection of neurons projecting to the injection site by the TVA AAV could allow direct retrograde infection of them by the RV, in the absence of G expression. The authors should redo these controls (e.g., omitting only the RV G AAV) in the four Cre lines used for the rest of the paper. This would be very easy to do and take little time but provide a much better set of controls.

2) Regarding cell-type specificity of their starter population, the references cited show nearly 100% co-localization of starter cell (TVA+ & RVG+) with Cre. For instance, Beier et al. (Cell 2015) Figure 1 and 1P shows that nearly all starter cells (yellow) are colocalized with TH. Those authors also used an anti-rabies glycoprotein to show specificity for RG expression. Watanabe-Uchida (Neuron 2012) also show that TVA expression overlaps with TH ~97%. The authors could could prove cell-type-specificity, for instance by performing immunos against one cell-type and rabies. They could show that RG expression is not leaky by injecting into wildtype mice. Perhaps better would be to inject both TC, RG and RVG into WT mice to demonstrate that they find no long-range projections.

3) To prove the cell type specificity of starter cells, as shown in all the references they mention, the authors need to do the same standard experiment. Immunostain for Chat in Chat-Cre transgenic mouse which is injected by AAV-FLEX-eGFP-2a-TVA, AAV-FLEX-RG and Rabies-dG-tdTomato+EnvA in HDB, then shows the percentage of co-localization of starter cell (yellow, eGFP-TVA positive & Rabies-tdTomato positive) with Chat immuno signal. Similar strategy for other cre lines. If the author can show a high percentage of colocalization, then it is done. But if the severe leaky of TVA cause a mismatch between starter cells and Chat immunosignal, then they need to prove the cell type specificity of glycoprotein G. To prove the cell type specificity of glycoprotein G it would be good to show that immunostaining signal for G is not in other cell types, but specifically in Chat neurons of Chat-cre mice and no signal in WT mice. Similar strategy for other Cre lines.

4) The concern about whether the variability of presynaptic neurons is correlated with the number of starter cells is not solved yet. There isn't much that can be done about this but it should be mentioned in the paper.

---

## [Author Response]

[Editors’ note: the author responses to the first round of peer review follow.]

*Do and colleagues perform a whole-brain mapping of both the inputs and outputs of the basal forebrain across its four major cell types using rabies virus. Understanding the connectivity of the basal forebrain is important and the new viral techniques used provide an unprecedented opportunity for revealing the cell-type-specific input-output logic of basal forebrain. These experiments required a lot of careful work and data analysis. The major take home message is that the inputs to all cell types are qualitatively similar but the output projections showed marked differences.*

*I was very enthusiastic about this paper until I got to supplementary Figure 1, the control experiment. What it shows is that the expression of rabiesdG-tdTomato+EnvA can occur without TVA or there is leaky expression of TVA without Cre (Figure 1). Figure 1 shows that about half of the rabiesdG-tdTomato+EnvA is not colocalized with AAV-TVA-GFP. These results indicate a severely leaky expression that causes non-specific infection of rabies in all cell types. The authors are to be commended for showing these controls. Clearly they realize their importance and try to focus on the relative fraction of inputs between cell-types. Unfortunately, however, having a mixed starter population can alone explain why the inputs of different cell types look the same, bringing the entire study into question.*

We appreciate the reviewer’s comment and apologize for not explaining this point clearly in our previous submission, which gave rise to the current confusion. The major concern is whether our input mapping was specific to each targeted cell type, given rabies infection of cells without Cre at the injection site (Figure 1—figure supplement 1). In fact, this issue is common to most of the published studies using rabies (RVdG)-based input tracing (Beier et al., 2015; Menegas et al., 2015; Miyamichi et al., 2013; Ogawa et al., 2014; Pollak Dorocic et al., 2014; Wall et al., 2013; Watabe-Uchida et al., 2012; Weissbourd et al., 2014), but the potential contamination of traced input neurons can be minimized as long as the results are interpreted based on proper control experiments, which are shown in our Figure 1—figure supplement 1. As demonstrated in previous studies, specificity of the starter cells (which must express both TVA and RG to allow rabies virus infection and transsynaptic spread) is ensured by the highly specific Cre/loxP recombination system (Watabe-Uchida et al., 2012). However, due to the extremely efficient interaction between TVA and EnvA-pseudotyped rabies virus, the very low-level expression of TVA in non-Cre-expressing cell types (not detectable based on fluorescent protein markers) allows rabies virus to infect these cells and label them with tdTomato at the injection site. Importantly, this issue does not compromise the mapping of long-range inputs because, due to the highly specific Cre/loxP recombination system, RG (required for transsynaptic spread of RV) is not expressed in any non-Cre-expressing cells at sufficient levels for trans-complementation of RVdG to allow transsynaptic spread of RV(Callaway and Luo, 2015; Miyamichi et al., 2013). Thus, while this rabies-based method should not be used for tracing local inputs near the injection site (excluded in our study), the standard practice for tracing long-range inputs is to determine the spatial extent of the local contamination using the control experiment without RG and exclude this region when analyzing the long-range inputs. This is exactly the purpose of the control experiments shown in our Figure 1—figure supplement 1, which are similar to or more rigorous than those performed by previous studies using the same approach (Beier et al., 2015; Pollak Dorocic et al., 2014; Wall et al., 2013; Weissbourd et al., 2014). We have now modified the text to explain this issue clearly (third paragraph of Results section).

*Other comments:*

*1) The authors report that subsets of ChAT cells in the basal forebrain may project to or receive projections from different brain regions. Figure 1 showed the targeting injection site is a bit posterior and lateral compared to the usual coordinates of HDB, which may lead to only partial infection of HDB. In addition, the infected ChAT and PV neurons shown in Figure 1 are significantly fewer in number than the total number of ChAT and PV neurons in HDB. The author should stain for ChAT, PV, SOM, vGlut2 and quantify whether the infection includes at least most of the HDB neurons or just part of it.*

For all the tracing experiments, we targeted the caudal portion of the BF, including the HDB, magnocellular preoptic nucleus, and substantia innominata (Materials and methods, subsection “Surgery and viral injections”). To address the reviewer’s question, we computed the average number of starter cells per brain sample for each cell type and compared it with the total cell number of that type within these BF nuclei (based on the in situ hybridization results from the Allen Mouse Brain Atlas). We found that the starter cell percentages are similar across the four cell types (ChAT, 16.3 ± 6.8%, mean ± s.e.m.; VGLUT2, 13.8 ± 3.1%; PV, 15.4 ± 1.7%; SOM, 17.6 ± 5.5%). As shown in the new Figure 1—figure supplement 2, the spatial distribution of starter cells was also similar across cell types. Note that although the infection does not include most of the BF neurons of the given type, which is common in RV tracing studies (see, e.g., Ogawa et al., 2014; Watabe-Uchida et al., 2012), it should not affect our main conclusion, which is based on the relative percentage of input from each brain area rather than the absolute number of inputs.

*2) A statement should be made with regard to whether the morphological analyses were made with the experimenter blinded to the treatment condition. This is especially important since all the quantification procedures from registration and detection module to cell counting are all manually performed or adjusted.*

The purpose of manual adjustment was to eliminate the errors produced by the registration module due to imperfect brain slice conditions. The majority of these analyses were performed by undergraduate research assistants who did not participate in the experimental design, had no knowledge of the whole-brain distributions of the traced inputs and outputs, and unaware of the conclusion of the study. In addition, signal extraction and manual adjustment were performed before the final quantification step (completely automatic), which counts the number of cells found within each brain structure, making it difficult to bias the result deliberately during the manual steps.

We have now included a statement on this in the Materials and methods (subsection “Registration module”).

*3) One particularly important piece of data in the paper is in Figure 2—supplement 1 showing physiological evidence for direct prefrontal inputs to cholinergic neurons. I think the authors should expand on these data and make into a full figure. There were not enough details provided to evaluate the experiment fully. The legend states the input is from "prefrontal cortex". Do the authors mean mPFC? Their rabies tracing data shows surprisingly little input from mPFC.*

We apologize for omitting some of the details in the original figure legend. The injection coordinate was 2.0 mm anterior to bregma, 1.5 mm lateral to midline and 2.0 mm from brain surface (Materials and methods, subsection “Slice recording”), which corresponds to orbital frontal/ insular areas of the PFC; this location was chosen based on our retrograde tracing results (Figure 2). We have now moved this result to the new Figure 4 and added more electrophysiology experiments.

*Reviewer #2:*

*In this study, the authors perform a detailed analysis of brain regions that input onto and receive outputs from four genetically defined cell-types in the basal forebrain. Using rabies-based labeling of monosynaptic inputs, they conclude that four major cell-types in the basal forebrain – PV+, SOM+, ChAT+, and VGLUT2^+^ neurons – receive qualitatively similar inputs, with striatum providing the main source of input. In addition, they report divergent outputs of each of these four cell-types as assessed by labeling of axons with an mCherry-expressing AAV, with ChAT+ neurons primarily projecting to the cortex, olfactory bulb, and hippocampus, PV+ neurons projecting to hypothalamus and pallidum, and both VGLUT2^+^ and SOM+ neurons forming reciprocal connections with the striatum and hypothalamus.*

*The analysis described in this study appear carefully done, and appropriate controls are shown. While the authors offer little in the way of interpretation, they have created a valuable, interactive resource that will be of great interest to other researchers interested in the basal forebrain.*

*There are several issue that should be addressed prior to publication. First, while the website is a valuable resource and welcome interactive supplement to the findings of the paper, there are some apparent discrepancies between the stated percentage values and the associated sunburst plot that either need to be addressed in the code or explained with better documentation or associated legends. For example, in the graphic showing inputs to ChAT cells, the "Basic cell groups and regions" lists 98.55% total inputs (it is understood from the legend of Figure 3—figure supplement 1 that some inputs come from unnamed sub-regions, though this could be mentioned in the main text). However, The 2nd level of the sunburst plot only includes "Brain stem" at 28.97% and "Cerebrum" at 64.80% – which sum to only 93.77%, not 98.44. As result, subsequent rings represent much more of the sunburst plot than the percentage would indicate – e.g. Inputs to ChAT neurons from Cerebral nuclei make up only 42.03% of the inputs, but well more than half of the sunburst plot. At a minimum this discrepancy needs to be better explained. Ideally, the sunburst plot would include an "un-named region" category, so that the graphic matches the percentage.*

We thank the reviewer for pointing out this apparent discrepancy. The visualization part of code was from the open source project of the Allen Institute for Brain Science. The mismatch between the section size and the actual value was due to the fact that there are unnamed regions in multiple brain structures. As the reviewer pointed out, the existence of unnamed region caused the percentage in parent structure to be larger than the sum of percentages from children structures in some cases (this was explained in the table legend of [Supplementary-material SD1-data]). The original visualization code calculated the sector size by summing the signals only from the named children structures and ignoring the un-named regions, which causes a mismatch in the actual percentage and the sector size. Since this representation is likely to cause confusion and apparent discrepancy as the reviewer pointed out, we have modified the code such that a white sector was included in each level of the sunburst to represent signals from un-named regions, and the sector size matches the percentage of each structure in the new version.

*Perhaps the most striking finding of the paper is the major input to all cell-types from the striatum. To my knowledge, a major input from the striatum to the basal forebrain has not been previously described. Could this be the result of hitting starter cells in the pallidum during the rabies-assisted retrograde tracing? Confirming functional connectivity between striatal projection neurons and basal forebrain neurons of all types, as in Figure 2 – supplement 1) would be very interesting and provide a necessary control to show that neurons of the BF proper also receive striatal input. This point needs to be addressed with electrophysiology and complete reconstruction as it would represent a radical change in our understanding of the BG wiring.*

Our finding that BF cholinergic neurons receive inputs from the ventral part of the striatum (ACB) is consistent with a previous ultrastructural tracing study (Zaborszky and Cullinan, 1992). To further confirm that the ACB indeed provides input to the BF rather than other pallidum regions (due to virus leakage into these regions in the rabies experiment), we injected Cre-dependent AAV expressing ChR2 into ACB of GAD2-Cre mice and measured light-evoked postsynaptic response in the caudal portion of the BF (including the HDB, magnocellular preoptic nucleus, and substantia innominata). We found that all four BF cell types receive GABAergic ACB inputs, and these results are now included in Figure 4.

*There is a great degree of variability in the number of labeled presynaptic neurons in different brains. Does this correlate in any way with the number of starter cells in each mouse? Greater analysis of the number and location of starter cells for each experiment would be useful.*

We have now performed further analyses of the number and location of starter cells in each sample (Figure 1—figure supplement 2, Figure 1—figure supplement 3; we excluded one of the VGLUT2 brain samples because many starter cells were found outside of the BF). We have also calculated the convergence index (the ratio between the number of inputs and starters, Figure 1—figure supplement 3), a value related to the transynaptic efficiency of the virus (Callaway et al., 2015). The variability in the convergence index was similar to other rabies tracing studies (Miyamichi et al., 2011; Miyamichi et al., 2014; DeNardo et al., 2015).

*Finally, the manuscript needs a greater discussion on the functional implications of the connectivity patterns discovered here, and the potential ways they do or do not meet expectations based on previous functional data. For example, the authors cite a recent paper showing PV+ BF neurons entrain cortical γ oscillations (Kim et al., 2015), but they show very little projection to the cortex from this population. The authors cite a recent paper (Hangya et al., 2015) that describes a recruitment of BF ChAT+ neurons by reinforcement with very long latency (~50 ms). Can the authors identify from their data an input that might explain this?*

The projection from BF PV+ GABAergic neurons to the cortex overall is indeed not as dense as the projections to the hypothalamus (Figure 3). In addition to the direct cortical projections, the entrainment of cortical γ oscillations shown by Kim et al. (2015) could also be partly mediated by PV projections to subcortical structures, which may in turn regulate cortical activity. We have now discussed this possibility in the revised text (Second paragraph of Discussion section).

We have also added a discussion on the functional implication of the inputs mapped in this study. In particular, Hangya et al. (2015) reported that BF cholinergic neurons exhibited strong responses to reinforcement signals in an auditory detection task. Our whole-brain mapping of the inputs to BF cholinergic neurons provides a list of candidate regions through which the reinforcement signals may be conveyed to these neurons. We have now added a discussion of this point in the revision (First paragraph of Discussion section).

*Reviewer #3:*

*Do et al. revealed input and output neural pathway to four different types of neurons (cholinergic, glutamatergic, parvalbumin (PV) and somatostain (SOM)) in the basal forebrain (BF) using Cre mice and transsynaptic retrograde tracing. They claimed that input pathway to four different cell types was similar but output pathway was different. Although the results were clear and informative to understand the physiological role of BF neurons, the following points need to be revised before publication.*

*Major points,*

*1) The author used cell type specific Cre recombinase-expressing mice (ChAT-Cre, VGLUT2-Cre, PV-Cre and SOM-Cre) to restrict starter cell for retrograde transsynaptic tracing. However, there is no information about specificity of Cre expression in each mouse line. How much Cre expressing neurons and which area of Cre expressing neurons were infected by rabies virus as starter cell should be provided.*

The four Cre mouse lines we used in the current experiment were shown to label each BF cell types with high specificity in a previous study (Xu et al., 2015), and this information was described in 1^st^ paragraph of our Results section (p. 5, lines 90-92). As shown in the control experiment (Figure 1—figure supplement 1), starter cells were found only in Cre mouse lines and not in wild-type controls. Together, these results indicate that the starter cells were restricted to each of the four cell types.

Based on the reviewer’s suggestion, we have provided a new figure summarizing the locations of the starter cells for each cell type (Figure 1—figure supplement 2), showing that the spatial distribution of starter cells was similar across cell types. We also computed the mean number of starter cells for each cell type and compared it with the total cell number of that type within these BF nuclei (based on the in situ hybridization results from the Allen Mouse Brain Atlas). We found that the starter cell percentages are similar across the four cell types (ChAT, 16.3 ± 6.8%, mean ± s.e.m.; VGLUT2, 13.8 ± 3.1%; PV, 15.4 ± 1.7%; SOM, 17.6 ± 5.5%). Note that although the infection does not include most of the BF neurons of the given type, which is common in RV tracing studies (Ogawa et al., 2014; Watabe-Uchida et al., 2012), it should not affect our main conclusion, which is based on the relative percentage of input from each brain area rather than the absolute number of inputs.

*2) Figure 2—figure supplement 1, the authors showed functional connection between prefrontal cortical (PFC) neurons and cholinergic neurons in BF. However, there is no information about feature of neurons in PFC labeled by retrograde transsynaptic tracing. In spite of this, the author used CaMKIIa promoter, which drives gene expression in glutamatergic neurons. Infected area of ChR2 expressing AAV in the PFC should be provided as well. Additionally, it is also important to show that brain area which is not innervate BF neurons from transsynaptic tracing is not functionally connected by expressing ChR2. This figure is referred only in the Discussion.*

The PFC neurons labeled by retrograde transsynaptic tracing from the BF are most likely to be glutamatergic (expressing CaMKIIα), since the great majority of cortical projection neurons are glutamatergic and this issue was specifically addressed in a previous study (Weissbourd et al., 2014). Based on the reviewer’s suggestion we have also included a figure showing a ChR2- EYFP expressing PFC region (Figure 4—figure supplement 1).

Regarding the brain areas not labeled with transsynaptic tracing, some of them may in fact provide inputs to the BF, but may be missed by RV-mediated tracing, as the RV tracing technique used in the current study does not label all presynaptic neurons (Marshel et al., 2010). While in future studies it would be interesting to determine whether some of the RV-negative brain areas provide inputs to the BF, it is beyond the scope of the current revision, given the vast number of brain areas that could be tested.

[Editors’ note: the author responses to the re-review follow.]

*1) The control experiments described in supplementary Figure 1 are good controls to do, and the ones in wild-type mice are fine (and important). For the controls in Cre mice, however, the authors use the wrong mouse line, for no apparent reason. The authors state that the use of GAD2-Cre mice "is likely to cause an overestimate of the exclusion zone"; this is plausible in the case of the PV-Cre and SOM-Cre lines, but not necessarily in the case of the Vglut2-Cre ones. It is known that EnvA-enveloped rabies virus can directly (non-transsynaptically) retrogradely infect TVA-expressing neurons projecting to an injection site (Huang…& Hantman 2013). In the worst case in the present study, direct retrograde infection of neurons projecting to the injection site by the TVA AAV could allow direct retrograde infection of them by the RV, in the absence of G expression. The authors should redo these controls (e.g., omitting only the RV G AAV) in the four Cre lines used for the rest of the paper. This would be very easy to do and take little time but provide a much better set of controls.*

We thank the reviewer for pointing out the issue with the mouse line used in the control experiments. We initially chose to use GAD2-Cre mice because GAD2+ neurons are the most abundant cell type in the BF, with a much higher density than all four cell types we used in the current study, which we thought would give a safer exclusion zone. We appreciate the reviewer’s concern and have added new control experiments using each of the four Cre lines and omitting only the RG in the injection. We found that non-specific labeling across all lines were largely restricted within the exclusion zone initially set using the GAD2-Cre mice. This result is now included in revised Figure 1—figure supplement 2.

*2) Regarding cell-type specificity of their starter population, the references cited show nearly 100% co-localization of starter cell (TVA+ & RVG+) with Cre. For instance, Beier et al. (Cell 2015) Figure 1 shows that nearly all starter cells (yellow) are colocalized with TH. Those authors also used an anti-rabies glycoprotein to show specificity for RG expression. Watanabe-Uchida (Neuron 2012) also show that TVA expression overlaps with TH ~97%. The authors could could prove cell-type-specificity, for instance by performing immunos against one cell-type and rabies. They could show that RG expression is not leaky by injecting into wildtype mice. Perhaps better would be to inject both TC, RG and RVG into WT mice to demonstrate that they find no long-range projections*

*3) To prove the cell type specificity of starter cells, as shown in all the references they mention, the authors need to do the same standard experiment. Immunostain for Chat in Chat-Cre transgenic mouse which is injected by AAV-FLEX-eGFP-2a-TVA, AAV-FLEX-RG and Rabies-dG-tdTomato+EnvA in HDB, then shows the percentage of co-localization of starter cell (yellow, eGFP-TVA positive & Rabies-tdTomato positive) with Chat immuno signal. Similar strategy for other cre lines. If the author can show a high percentage of colocalization, then it is done. But if the severe leaky of TVA cause a mismatch between starter cells and Chat immunosignal, then they need to prove the cell type specificity of glycoprotein G. To prove the cell type specificity of glycoprotein G it would be good to show that immunostaining signal for G is not in other cell types, but specifically in Chat neurons of Chat-cre mice and no signal in WT mice. Similar strategy for other Cre lines.*

We thank the reviewer for the excellent suggestions. To test the specificity of RG expression, we performed immunostaining of RG in mice expressing tdTomato or mCherry in Cre+ cells (by crossing tdTomato reporter line with a Cre driver line or by injecting AAV2-EF1α-FLEX-mCherry into a Cre line). We found high colocalization between RG and tdTomato/mCherry expression (Figure 1—figure supplement 1). In contrast, in wild type mice lacking Cre-recombinase expression, we found no RG labeling. Since RG is required for transsynaptic labeling, the high specificity of RG expression ensures specificity of the starter cell population. Furthermore, wild type mice injected with TVA, RG and RV showed no long-range projections (Figure 1—figure supplement 2). Collectively, these experiments demonstrate the cell type specificity of the starter population.

*4) The concern about whether the variability of presynaptic neurons is correlated with the number of starter cells is not solved yet. There isn't much that can be done about this but it should be mentioned in the paper.*

We measured the convergence index (ratio between the number of input cells and starter cells) and found it to range between 4.3 and 77.7 for the four cell types (Figure 1—figure supplement 4). Such a level of variability was also found in some other tracing studies using similar methods (Miyamichi et al., 2011, DeNardo et al., 2015). We have mentioned this in the revised manuscript (third paragraph of Results).